# mTOR Inhibitors Modulate the Physical Properties of 3D Spheroids Derived from H9c2 Cells

**DOI:** 10.3390/ijms241411459

**Published:** 2023-07-14

**Authors:** Megumi Watanabe, Toshiyuki Yano, Tatsuya Sato, Araya Umetsu, Megumi Higashide, Masato Furuhashi, Hiroshi Ohguro

**Affiliations:** 1Department of Ophthalmology, School of Medicine, Sapporo Medical University, Sapporo 060-8556, Japan; watanabe@sapmed.ac.jp (M.W.); araya.umetsu@sapmed.ac.jp (A.U.); megumi.h@sapmed.ac.jp (M.H.); 2Department of Cardiovascular, Renal and Metabolic Medicine, Sapporo Medical University, Sapporo 060-8556, Japan; oltomwaits55@gmail.com (T.Y.); satatsu.bear@gmail.com (T.S.); furuhasi@sapmed.ac.jp (M.F.); 3Department of Cellular Physiology and Signal Transduction, Sapporo Medical University, Sapporo 060-8556, Japan

**Keywords:** H9c2, 3D spheroid culture, 2D culture, mTOR, rapamycin, Seahorse Bioanalyzer

## Abstract

To establish an appropriate in vitro model for the local environment of cardiomyocytes, three-dimensional (3D) spheroids derived from H9c2 cardiomyoblasts were prepared, and their morphological, biophysical phase contrast and biochemical characteristics were evaluated. The 3D H9c2 spheroids were successfully obtained, the sizes of the spheroids decreased, and they became stiffer during 3–4 days. In contrast to the cell multiplication that occurs in conventional 2D planar cell cultures, the 3D H9c2 spheroids developed into a more mature form without any cell multiplication being detected. qPCR analyses of the 3D H9c2 spheroids indicated that the production of collagen4 (COL4) and fibronectin (FN), connexin43 (CX43), β-catenin, N-cadherin, STAT3, and HIF1 molecules had increased and that the production of COL6 and α-smooth muscle actin (α-SMA) molecules had decreased as compared to 2D cultured cells. In addition, treatment with rapamycin (Rapa), an mTOR complex (mTORC) 1 inhibitor, and Torin 1, an mTORC1/2 inhibitor, resulted in significantly decreased cell densities of the 2D cultured H9c2 cells, but the size and stiffness of the H9c2 cells within the 3D spheroids were reduced with the gene expressions of several of the above several factors being reduced. The metabolic responses to mTOR modulators were also different between the 2D and 3D cultures. These results suggest that as unique aspects of the local environments of the 3D spheroids, the spontaneous expression of GJ-related molecules and hypoxia within the core may be associated with their maturation, suggesting that this may become a useful in vitro model that replicates the local environment of cardiomyocytes.

## 1. Introduction

The heart requires high levels of energy. Because of this, the metabolic characteristics of cells from this organ are different from other myocytes as well as non-myocytes [1]. It has been shown that the matured cardiomyocytes from adult mammalian hearts obtain most of their energy from fatty acid oxidation rather than glycolysis [2,3]. Upon aging and/or cardiac stress, the cellular metabolism of myocardial tissues, including cardiomyocytes, is altered and this can induce several functional defects of the heart [4,5]. In fact, in heart failure, the cardiomyocytes attempt to use more glucose for ATP production instead of fatty acids. This is similar to the cellular metabolic pattern observed in the fetal cardiomyocytes [2], suggesting that such an alteration in the metabolism pattern may contribute to the senescence of cardiomyocytes as well as cardiac aging [6]. In addition, oxygen (O_2_) supply is extremely important for the physiological functions of the heart, and in fact, insufficient O_2_ supply following a myocardial infarction or other etiology typically induces heart failure [7]. For the purpose of elucidating organotypic functions under physiological and pathological conditions for the heart, a straightforward in vitro approach using cardiac cells directly isolated from living tissue could be considered to be a native state, instead of animal models [8]. In the standardized two-dimensional (2D) cell culture method, cells are cultured on the planar surface of plastic plates, dishes, and flasks and the expanding cell lines are observed, thus allowing cell status to be monitored by examination of the bioproducts. However, although such 2D cultured cells are initially expected to similarly respond with an in vivo system for several chemicals, including drugs, toxins, signaling modifiers and others, we have come to the realization (based on recent investigations) that such systems replicate similar but also different aspects from actual in vivo properties [8,9,10,11]. Therefore, to overcome such disadvantages that are associated with the conventional 2D planar culture system, several in vitro three-dimensional (3D) culture models using cardio-related cells that mimic the heart have been actively developed [12,13,14]. 

H9c2 cardiomyoblasts were originally derived from embryonic rat ventricular tissue and are the most extensively characterized cell line used in cardiomyocyte research. Although H9c2 cells have a limited capacity to beat by themselves, there are similarities in gene expression patterns, plasma membrane morphology, hypertrophic responses, and cytoprotective signaling, including the Akt/mammalian target of the rapamycin (mTOR) signaling pathway between H9c2 cells and primary cardiomyocytes [15,16,17,18,19,20,21]. Thus, H9c2 cells are generally thought to be an acceptable alternative for primary cardiomyocytes; data regarding the properties of H9c2 cells in the 3D culture models are limited. As a result, although we recently analyzed the protective roles of mTOR inhibition in cardiomyocyte necrosis through the modulation of autophagic flux and the regulated cell death pathway [22,23], the issue of whether this is the case in the 3D culture cardiomyocytes remains unclear. This is a critical issue, since there are clearly differences in Akt/mTOR signaling between the 2D and 3D cultures of cancer cell lines [24].

We independently developed in vitro 3D spheroid models that replicate diseases that affect orbital fatty tissues in which adipocytes were grown within a 3D conic space, such as Graves’ orbitopathy (GO). Using 3D drop-culture methods, we were able to successfully produce 3D spheroids from human orbital fibroblasts (HOFs) in patients with GO, and our findings indicate that HIF2A played a pivotal role in the LOX-dependent accumulation of ECM molecules [25]. Furthermore, to elucidate the molecular pathology of the prostaglandin (PGs)-induced deepening of the upper eyelid sulcus (DUES), we also employed this 3D culture system using HOFs and 3T3L1 cells, and found that PGs significantly suppressed the sizes of the 3D spheroids, and modulated the spatial distribution of several ECMs that surround the 3D spheroids [26,27]. Based upon these collective findings, we propose that our developed 3D cell culture would be viable in vitro models of GO and DUES and would also be a promising strategy for establishing in vitro models that mimic various pathogenic states using their target cells. In fact, following these HOFs and 3T3-L1 preadipocytes, we also succeeded in obtaining various in vitro 3D spheroid models using several types of non-cancerous cells, including fibroblasts from various sources [25,28,29,30], human trabecular meshwork cells [31,32,33], and cancerous cells, including the A549 lung adenocarcinoma cell [34] and various malignant melanoma cell lines [35].

Therefore, in the current study, to establish a physiologically relevant in vitro cardiac 3D spheroid model, our recently developed 3D drop spheroid culture method was applied using H9c2 cells, and the morphology, physical properties, and cellular metabolisms of the 3D spheroids, as well as the effects by mTOR inhibitors, were evaluated.

## 2. Results

### 2.1. Establishment of the In Vitro 3D H9c2 Spheroid Model and Its Characterization

To establish an in vitro model that replicates the physiological cardiac muscle environments, 3D spheroids of H9c2 cells were generated by our recently developed 3D hanging drop culture methods using several non-cancerous cells [25,27,28,29,30,31,32,33,36] and cancerous cells [34,35]. To study the growth process of the 3D H9c2 spheroids, the changes in their appearance and physical properties, including sizes and stiffness, were evaluated over 4 days. As shown in Figure 1, phase contrast microscopy (PC) images indicated that within a 28 μL 3D spheroid culture medium, 20,000 H9c2 cells had been dispersed and the cells started to coalesce at 3 h after the seeding, with premature spheroids being formed by 6–12 h. On Day 1, a round-shaped 3D spheroid had been formed. Thereafter, their sizes became gradually smaller until Day 4 (Figure 2A), and their stiffness constantly increased, reaching a plateau by Day 3 (Figure 2B). Scanning electron microscopy analysis was performed to investigate their ultrastructure. As shown in Figure 3, lower magnification images (×150) demonstrated that the 3D H9c2 3D spheroids became smaller until Day 3, in agreement with PC images. However, the H9c2 3D spheroids by Day 4 were smaller and deformed with shapes that were similar to a rugby ball. Furthermore, higher magnification images (×10^4^) showed that the ECM deposits distributed on the 3D spheroid surface were increased until Day 2 and then gradually diminished until Day 4. To confirm that these down-sizing and stiffening processes were not artifacts or cell death within the inside of the 3D spheroid, DAPI nuclei staining was preformed and the total cell numbers of the 3D spheroids were determined (Figure 4). As shown in Table 1, the numbers of cells within a 3D spheroid at Day 3 (19,843 ± 2002 cells) were nearly identical to those that were initially harvested (approximately 20,000 cells). Therefore, the total cell numbers within the 3D H9c2 spheroids did not increase until at least Day 3. Furthermore, to estimate inter cellular binding properties within the 3D spheroids, their trypsin-induced dispersion was compared with 2D planar cultured cells. As shown in Figure 5, upon treatment with 0.05% trypsin, the 2D planar cultured H9c2 cells were dispersed within 3 min, but 5 h were required for the 3D H9c2 spheroids to disperse. 

Therefore, these collective results indicate that the natures of the 3D H9c2 spheroids were different from those of the 2D planar cultured H9c2 cells. That is, the H9c2 cells did not multiply in our 3D spheroid culture system as was observed in the conventional 2D planar cell cultures, but the 3D spheroids matured during the course of the drop 3D spheroid cultures. 

### 2.2. Effects of mTOR Inhibitors on the Physical Properties of 3D H9c2 Spheroids

To study the effects of mTOR inhibitors on the in vitro 3D spheroid H9c2 model, the physical properties, mean sizes, and physical stiffness analysis were evaluated in the absence and presence of mTORC1 inhibitors. There are two mTOR complexes that have distinct functions. Rapa is an allosteric inhibitor for the mTOR complex 1 (mTORC1), whereas Torin 1 is an ATP-competitive mTOR inhibitor that suppresses the action of both mTORC1 and mTOR complex 2 (mTORC2). Successful inhibition of the mTOR complexes by Rapa and Torin 1 was confirmed by immunocytochemical analyses: Rapa suppressed S6 phosphorylation but not Akt phosphorylation, whereas Torin 1 inhibited the phosphorylation of S6 and Akt (Figure 6 and Figure 7). To elucidate the effects of mTOR inhibitors on the physical characteristics of the 3D H9c2 spheroids, their sizes and stiffness were compared among non-treated controls and spheroids that had been treated with a 10 nM mTOR inhibitor (Rapa or Torin 1) or a 5 nM mTOR stimulator (MHY-1485). As shown in Figure 8, Torin 1 induced a significant downsizing of the H9c2 spheroids during Day 2 and Day 3, and Rapa also induced a modest reduction in the size of 3D spheroids despite that fact there were no significant effects by MHY-1485. In addition, the stiffness of the 3D H9c2 spheroids was also substantially decreased by Torin 1 and tended to be decreased by Rapa at Day 3, suggesting that the inhibition of both mTOR complexes induced substantial effects toward the physical aspects of the 3D H9c2 spheroids and mTORC1 inhibition alone also played a role in this process. Since it is well known that mTOR inhibitors reduce cell sizes as well as cell densities [37], such effects of mTOR inhibitors could be caused by reducing cell sizes, cell densities, or both. To examine these possibilities, the density of 2D cultured cells and total cells within 3D spheroids were compared among these four conditions. As shown in Table 1, the cell densities (cells/0.1 mm^2^) of 2D cultured cells during 3 days of Rapa (20.25 ± 4.19) or Torin 1 (8.75 ± 4.19) treatment were significantly lower than the non-treated control (36.0 ± 10.98) or MHY-1485 (26.0 ± 1.15). On the other hand, the total cell numbers of the 3D spheroids were essentially unchanged (non-treated control, 19,843 ± 2002; Rapa, 17,637 ± 2622, Torin 1, 18,707 ± 2725, WHY-1485, 20,449 ± 2891). Therefore, those results suggest that the down-sizing of the 3D H9c2 spheroids by mTORC1/2 inhibition can be attributed to a reduction in each cell volume rather than to the number of cells. 

### 2.3. Effects of mTOR Inhibitors on Cellular Metabolic Functions of 2D and 3D Cultured H9c2

To further study the effects of mTOR inhibitors toward 2D and 3D cultured H9c2 cells, real-time cellular metabolic functions were measured using a Seahorse Bioanalyzer (Figure 9A–D). The ratio of mitochondrial maximal respiration to basal OCR, which reflects spare respiratory capacity, one of the indicators of mitochondrial function, was comparable between the 2D and 3D cultures, although the experimental protocols for real-time metabolic analyses are different for 2D cells and 3D spheroids (Figure 9E). However, interestingly, in the case of the 3D H9c2 spheroids, there were significant differences in basal OCR and ATP-linked respiration between treatment with the mTOR activator MHY and treatment with the mTOR inhibitor Torin 1, whereas no difference in response to mTOR modulators was observed for the indices of mitochondrial respiration, and glycolytic capacity was somewhat decreased by the treatment with MHY (Figure 9F,G). These results indicate that H9c2 cells cultured in 2D and 3D configurations exhibit different cellular metabolic responses to mTOR signaling.

### 2.4. Effects of mTOR Inhibitors on the mRNA Expression of ECM Proteins, Gap Junction-Related Molecules, and HIF1 of 2D and 3D Cultured H9c2

To determine the kinds of underlying mechanisms that are involved in the effects that are induced in the 2D and 3D cultured H9c2 cells by the mTOR inhibitors as shown above, as possible related molecules, the mRNA expression of major ECM proteins including collagen (COL)1, 4, and 6, fibronectin (FN), STAT3, and α-smooth muscle actin (α-SMA), gap junction-related molecules including ZO-1, β-catenin and N-cadherin, and HIF-1 were evaluated. In a comparison between 2D cultured H9c2 cells and 3D H9c2 spheroids, (1) significant differences were found for the expression of ECM proteins between 2D and 3D cultured H9c2 cells, and substantially higher expressions of COL4 and FN and lower expressions of COL6 and α-SMA were observed in the 3D H9c2 spheroids as compared with 2D H9c2 cells. (2) Among gap junction-related molecules, mRNA expressions of Cx43, β-catenin, and N-cadherin were markedly elevated in 3D H9c2 spheroids, and as a representative molecule among them, a significant higher expression of N-cadherin was also confirmed using Western blot analysis (Appendix A). (3) Higher expressions of STAT3 and HIF1 were also found for the 3D H9c2 spheroids (Figure 10). Since STAT3 was recently identified as the master regulator for inducing 3D spheroid architecture [36], and HIF1 is recognized as the major transcription factor that is activated in response to hypoxia [38], these collective results rationally suggest that our currently developed in vitro 3D H9c2 spheroid model could replicate quite different local environments as compared with conventional 2D planar culture models. However, in contrast, in the presence of an mTOR inhibitor, Rapa or Torin 1, or an mTOR stimulator, MHY-1485, the mRNA expressions of those molecules were not significantly different between the 2D and 3D cultures, except for: (1) ECM proteins; Rapa (2D: down-regulation *COL1* and *COL6*, 3D: down-regulation of *COL1*, *COL4*, and *FN*, up-regulation of *COL6*) and MYH-1485 (2D: down-regulation of *α-SMA*, 3D: down-regulation of all five ECM proteins) (Figure 11); (2) gap junction-related molecules; Rapa (2D: up-regulation of *N-cadherin*, 3D: down-regulation of *β-catenin*) and Torin 1 (3D: down-regulation of *β-catenin*); (3) STAT3: Rapa (2D: down-regulation); (4) HIF1; Rapa (2D and 3D: down-regulation), Torin 1 (2D: down-regulation), and MHY-1485 (2D: down-regulation, 3D: up-regulation) (Figure 12). 

## 3. Discussion

In the current investigation, 3D H9c2 spheroids were successfully produced by the same experimental protocol using a 384-hanging drop array plate as described in our recent studies using mouse preadipocyte 3T3-L1 cells [27] as well as various other cells. The formation of 3D H9c2 spheroids and their subsequent maturation, as shown in Figure 1, closely resembled that of 3T3-L1 cells [27]. Interestingly, the results of the qPCR analyses suggested that GJ-related molecules, which are essentially required for the cardiac functional syncytium [39], were spontaneously expressed upon 3D H9c2 spheroid formation. In addition, since it has been shown that an O_2_ gradient exists within the 3D spheroids, similar to the cardiac wall, the significantly higher expression of the HIF1 gene may also suggest the existence of such an O_2_ gradient within our 3D H9c2 spheroid, although changes in protein levels of HIF1 and its downstream protein were not measured. In addition, mTOR modulators induced different effects toward cellular proliferations as well as the cellular metabolic states of 2D and 3D cultured H9c2 cells. Therefore, these collective current results rationally suggest that our newly developed in vitro 3D H9c2 spheroid model could be more applicable than conventional 2D cultured models for use in research in the fields of cardiac physiology as well as pathology. 

The use of 3D spheroid cell cultures has recently received great attention for use in establishing suitable in vivo models for a variety of physiological and pathological conditions [40]. In fact, compared with the conventional 2D planar cell culture, the nature of the intercellular interaction for 3D spheroids is different; that is, each cell can, at any location, interact with another cell within the 3D spheroid similar to those of in vivo organs. In contrast, side-by-side intercellular interactions occur in the 2D cultures. In addition, the protein networks, including several ECM proteins, cell junction proteins, and other surrounding cells within 3D spheroids, may also be similar to those of in vivo organs but different from those of the 2D cultured cells. Thus, it would therefore appear that such 3D spheroids replicate real tissues and organs in terms of their biological characteristics [41]. However, it should be noted that such advantageous aspects in the 3D spheroids may greatly differ depending on the methods used for their preparation. Several methods for 3D cell culture have been developed, and they can be roughly classified into scaffold-assisted or non-scaffold-assisted methods [42]. In the non-scaffold-assisted methods, several different techniques to prepare 3D spheroids, including pellet cultures, liquid overlay, hanging drop, spinner cultures, rotating wall vessels, microfluidics, and magnetic levitation, have been used [42]. Among these methods, we focused on a modified hanging drop 3D spheroid culture using a 384-hanging drop array plate [43,44] because we paid a great deal of attention to the fact that 3D spheroids could be spontaneously obtained within the culture medium droplet. In fact, in our previous report, dispersed 3T3-L1 cells within the culture medium droplet spontaneously became associated with one another within a few hours to form premature 3D spheroids, which then gradually grew into their matured form [27]. In addition, such formed 3D 3T3-L1 spheroids demonstrated quite different biological activities as compared with 2D cultured 3T3-L-1 cells even though the culture conditions were exactly the same for both systems, except that the culture plates were different [36]. 

The Akt-mTOR signaling pathway is important in regulating the cell cycle and is essential in promoting the growth, proliferation, and differentiation of adult stem cells [45]. The activation of Akt has vital effects on cardiomyocytes, including increasing cell size, inhibiting apoptosis, and altering glucose metabolism [46]. In the present study, we found that mTOR inhibitors, Rapa, and Torin 1 induced different effects on 2D H9c2 cells and 3D H9c2 spheroids; that is, mTORC1/2 inhibition suppressed cell multiplication in the former, but in the latter, it caused the down-sizing of each cell within the 3D spheroid without having an effect on cell multiplication. At the time of writing, we do not know why mTOR inhibition-induced effects were different between 2D culture cells and 3D spheroids. One possibility might be related to the O_2_ gradient within the 3D spheroid. Since O_2_ is a well-recognized and important factor in maintaining cellular energy homeostasis via mitochondria for cell proliferation and growth, hypoxia greatly deteriorates cellular metabolism as well as physiological reactions and, in turn, to adapt to these conditions, hypoxia-inducible gene expression, metabolism, reactive oxygen species, and autophagy are all stimulated. In addition, these hypoxia-related signaling events are closely linked with the cellular energy and nutrient-sensing pathways; that is, AMP-activated protein kinase (AMPK) and mTORC1 [47].

In addition to the cellular properties of the matured cardiomyocytes, 3D H9c2 spheroids may also better mimic the developmental process of the cardiogenesis compared to 2D culture models because it is well known that the embryonic or fetal heart develops under physiologically hypoxic concentrations. Indeed, recent studies have demonstrated that HIF1 signaling, which was suggested to be activated in 3D H9c2 spheroids, plays an important role in early cardiogenesis [48,49,50]. In addition, it has been suggested that metabolic adaptation through mTOR signaling plays an essential role in cardiac development [51]. Taking these collective issues into account, 3D cultured H9c2 spheroids, in which an O_2_ gradient is present and metabolic responses by mTOR signaling clearly occur, may be a more suitable model for studying developing cardiac cells than 2D planar cultures of H9c2 cells.

Although the possible mechanisms responsible for causing these differences between 2D and 3D cultures of H9c2 have not been elucidated at this time, several diverse biological properties between them have been identified, suggesting that our newly developed 3D H9c2 spheroid may be a useful in vitro model for understanding cardiac pathophysiology.

## 4. Materials and Methods

### 4.1. 2D Planar and 3D Spheroid Cultures of H9c2 Cells

The 2D planar cultures of the H9c2 cells, rat cardiomyoblast cells obtained from the ATCC (American Type Culture Collection), were prepared as described in a previous report [22]. Thereafter, they were further used to prepare 2D cultures or 3D spheroid cultures as described in our previous studies using human orbital fibroblasts [25,26], 3T3-L1 preadipocytes [27], and other cells [29,32] during 4 days in the absence or presence of 10 nM mTOR inhibitor, rapamycin (Rapa), or Torin 1 or 5 μM mTOR activator, MHY-1485. The concentrations of mTOR inhibitors and mTOR activators that were administered were the same as in previous reports [22,52].

### 4.2. Physical Properties Measurements of the 3D HTM Spheroid

The configuration of the 3D H9c2 spheroids was observed by phase contrast (PC) microscopy and scanning electron microscopy, methods that have been described in our previous reports [27,53]. For evaluation of the mean size (μm) and stiffness (required force to compress semidiameter, μN/μm) of each 3D spheroid, we used PC images and a micro-squeezer as described in our previous reports [25,26].

### 4.3. Immunocytochemistry of 2D Cultures H9c2 Cells and 3D H9c2 Spheroids

Immunocytochemistry of the 2D cultured H9c2 cells and 3D H9c2 spheroids was processed as described in our previous reports [26,27] using 1: 200 dilutions of first antibody including an anti-human S6 ribosomal protein (5G10) rabbit mAb (#2217, Cell Signaling Technology, Danvers, MA, USA), phosphor-S6 ribosomal protein (Ser235/236) antibody (#2211, Cell Signaling Technology, Danvers, MA, USA), Akt antibody (#9272, Cell Signaling Technology, Danvers, MA, USA) or phosphor-Akt (Ser473) (#9271, Cell Signaling Technology, Danvers, MA, USA), 1:1000 dilutions of a goat anti-rabbit IgG (488 nm) with phalloidin (594 nm), and DAPI. Thereafter, confocal immunofluorescent images of 2D H9c2 cells and 3D spheroid were obtained as described in our previous reports [26,27].

### 4.4. Measurement of Real-Time Cellular Metabolic Functions

The oxygen consumption rate (OCR) and the extracellular acidification rate (ECAR) of the 2D and 3D cultured H9c2 cells that were treated with or without mTOR inhibitors, 10 nM Rapa or 10 nM Torin 1, or an mTOR stimulator 5 μM, MHY-1485 (MHY), were measured using a Seahorse XFe96 real-time metabolic analyzer (Agilent Technologies, Santa Clara, CA, USA.) according to the manufacturer’s instructions.

For the 2D planar cultures of cells, 2.0 × 10^4^ 2D cultured H9c2 cells treated as above were placed in each well of an XFe96 Cell Culture Microplate (Agilent Technologies, #103794-100) and incubated at 37 °C. At the time of the assay, the culture medium was replaced with pre-warmed Seahorse XF DMEM assay medium (pH 7.4, Agilent Technologies, #103575-100) containing 5.5 mM glucose, 2.0 mM glutamine, and 1.0 mM sodium pyruvate. For the 3D spheroids, each spheroid was placed in accordance with the User Guide for the Agilent Seahorse XFe96 Spheroid Microplate and Flux Pak (Agilent Technologies #102905-100). Briefly, to coat the plate with Cell-Tak™ (#354240, Corning, New York, NY, USA) before placing the spheroids in it, 200 μL of 2 mg/mL in 5% acetic acid was added in 2.8 mL of 0.1 M sodium bicarbonate, and 30 μL of this Cell-Tak Mix was then placed to each well of an XFe96 Spheroid Microplate followed by incubation for 1 h in a non-CO_2_ incubator at 37 °C. The Cell-Tak Mix was then aspirated from the plate, and the plate was washed twice with 400 µL of sterile 37 °C water and allowed to air dry. A pre-warmed 175 μL of Seahorse XF DMEM assay medium (pH 7.4, Agilent Technologies, #103575-100) containing 5.5 mM glucose, 2.0 mM glutamine, and 1.0 mM sodium pyruvate was added to each well of the XFe96 Spheroid Microplate coated with Cell-Tak Mix, and individual spheroids from the culture well were placed in each well of the Agilent Seahorse XFe96 Spheroid Microplate containing the prewarmed assay medium. The assay plates for 2D cells and 3D spheroids were then incubated in a CO_2_-free incubator at 37 °C for 1 h prior to the measurements. OCR and ECAR were simultaneously measured using an XFe96 extracellular flux analyzer (Agilent Technologies, Santa Clara, CA, USA) at the baseline and under the following sequential injections of 2.0 μM oligomycin, 5.0 μM carbonyl cyanide-p-trifluoromethoxyphenylhydrazone (FCCP), a mixture of 1.0 μM rotenone and 1.0 μM antimycin A, and 10 mM 2-deoxyglucose. Since the sensitivity of sequential drug injections is different between 2D and 3D conditions, 3 cycles of each measurement were employed for 2D cells, 8 cycles for measurement with oligomycin, and 4 cycles for other measurements were employed for the 3D spheroids. The OCR and ECAR values were normalized to the amount of protein assessed by a BCA protein assay (TaKaRa BCA Protein Assay) per well in 2D cells and to the number of spheroids in 3D spheroids.

### 4.5. Assays for Trypsin-Induced Dispersion

To compare the trypsin-induced destruction of the 2D and 3D cultured H9c2 cells, the period required for them to disperse in the presence of a 0.05% trypsin solution was determined using phase contrast microscopy.

### 4.6. Measurements of Cell Density of 2D Cultured H9c2 Cells and Total Cell Numbers within a Single 3D H9c2 Spheroid

For measurements of the 2D cell density (cell numbers/0.1 mm^2^), 2D cultures of H9c2 cells were processed as above using a 6-well culture dish over 3 days. After washing twice with phosphate-buffered saline (PBS), cells within each well were recovered by 0.25% trypsin dispersion and subsequent centrifugation. Then, each cell pellet was resuspended within 100 μL PBS and, thereafter, 10 μL aliquot was subjected to hemocytometer measurements.

Alternatively, for the estimation of total cell numbers within a 3D spheroid, since H9c2 cells were concentrically arranged from the center to the surface within the 3D HTM spheroid in the DAPI staining image, the numbers of cells within a 3D spheroid were evaluated as follows: (1) the volumes of a 3D spheroid and a representative cell were calculated by assuming their tentative diameters estimated by the largest cross-section of phalloidin images of the 3D spheroid (*n* = 5) and the distance between two adjacent nuclei stained by DAPI (*n* = 5 for one section and was repeated five times using different preparations), respectively; and 2) the cell numbers within the 3D spheroid were estimated by division of the volume of the overall 3D spheroid with the volumes of the cells within it.

### 4.7. Other Methods

Quantitative PCR using specific primers (Appendix A) and statistical analyses using the Graph Pad Prism 8 (GraphPad Software, San Diego, CA, USA) were performed as described in a previous report [27]. Western blotting analysis using cell lysates from 2D and 3D cultured H9c2 cells were performed as described in our recent study [22] using a specific polyclonal antibody against N-cadherin or α-tubulin. For the estimation of the statistical difference between study groups, one-way ANOVA was used followed by a Tukey’s multiple comparison test.

## Figures and Tables

**Figure 1 ijms-24-11459-f001:**
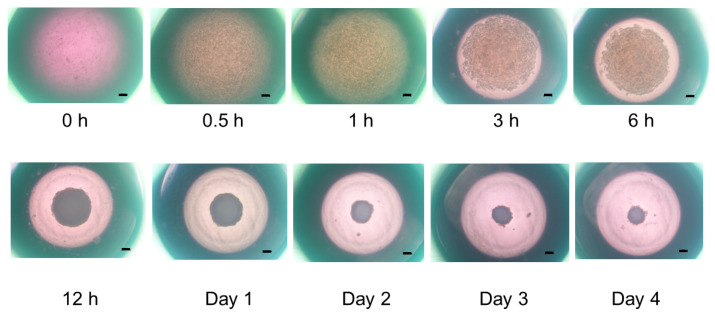
Maturation process of the 3D H9c2 spheroid. To demonstrate the maturation process of the 3D H9c2 spheroids, representative phase contrast microscopy (PC) images of the 3D H9c2 spheroids were collected at several time points (0, 0.5, 1, 3, 6, or 12 h, or Day 1, 2, 3, or 4) (Scale bar: 100 µm).

**Figure 2 ijms-24-11459-f002:**
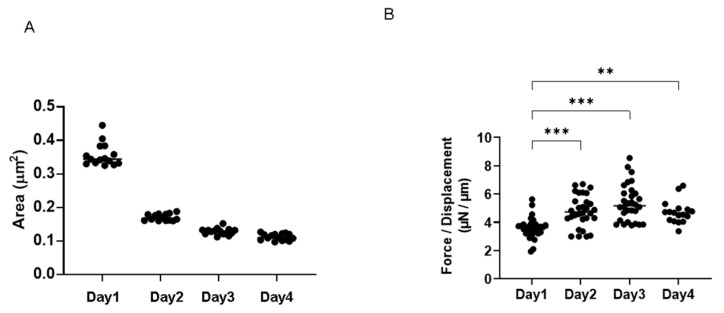
Physical aspects of the 3D H9c2 spheroids. Regarding the physical characteristics of the H9c2 spheroids, their mean sizes were measured at several time points (Day 1, 2, 3, or 4) and plotted in Panel (**A**). Alternatively, as the physical stiffness, the force (μN) required to induce deformation until their half diameter (μm) was reached was measured at several time points (Day 1, 2, 3, or 4) using a micro-squeezer, and the force/displacement (μN/μm) was potted in Panel (**B**). ** *p* < 0.01, *** *p* < 0.005, and other pairs were not statistically significant.

**Figure 3 ijms-24-11459-f003:**
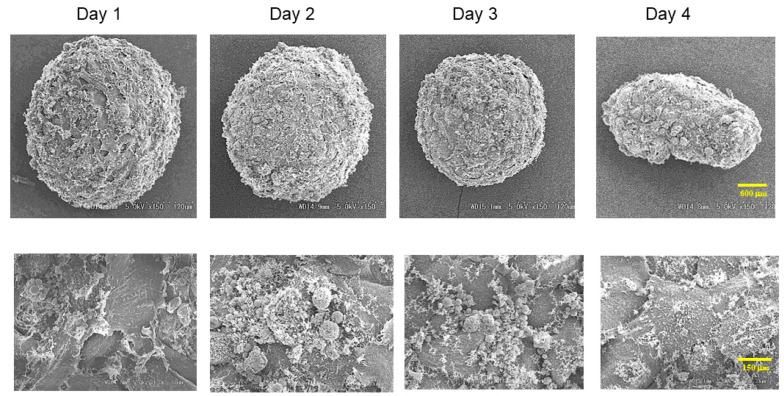
Ultrastructure of the H9c2 spheroids. To observe the ultrastructure of the H9c2 spheroids, representative scanning Electron microscope (SEM) images at several time points (Day 1, 2, 3, or 4) are shown (upper; low magnification, Scale bar: 600 µm, lower; high magnification, Scale bar: 120 µm).

**Figure 4 ijms-24-11459-f004:**
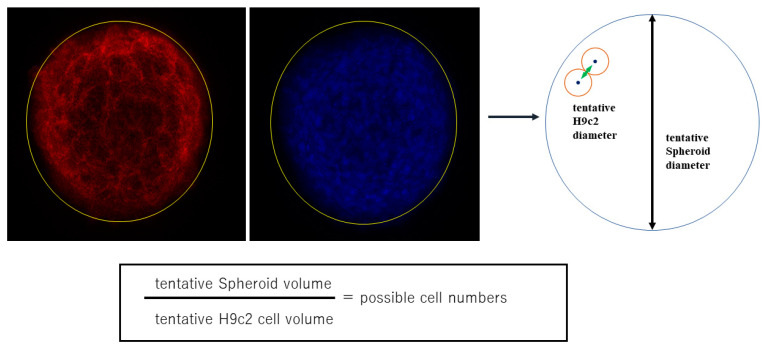
Estimation of cell numbers within a single 3D H9c2 spheroid. H9c2 spheroids were stained with DAPI (blue) and phalloidin (red). The largest cross-section of phalloidin images of the 3D spheroid (*n* = 5) was selected among confocal microscopy images, and their diameters were measured (tentative 3D spheroid diameter). Alternatively, the distance between two adjacent nuclei stained by DAPI (*n* = 5 for one section and repeated five times using different preparations) was measured (tentative H9c2 cell diameter). Based upon these diameters, the cell numbers contained within the 3D spheroid were estimated by division of the volume of the overall 3D spheroid with the volumes of the cells within it.

**Figure 5 ijms-24-11459-f005:**
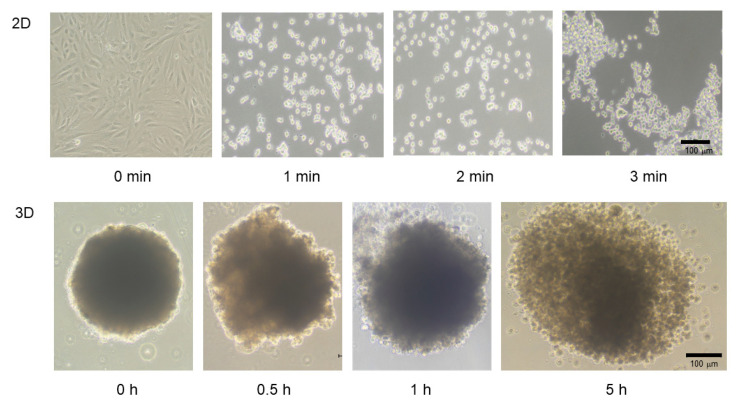
Trypsin-induced dispersion of 2D planar cultured or 3D spheroid cultured H9c2 cells. The 2D planar cultured or 3D spheroid cultured H9c2 cells were treated with 0.05% trypsin for 3 min or 5 h, respectively. Representative phase contrast microscopy images of 2D cells and 3D spheroids are shown in panels, respectively (scale bar; 100 μm). Experiments were repeated in triplicate using fresh preparations (2D; *n* = 5, 3D; *n* = 10 spheroids each).

**Figure 6 ijms-24-11459-f006:**
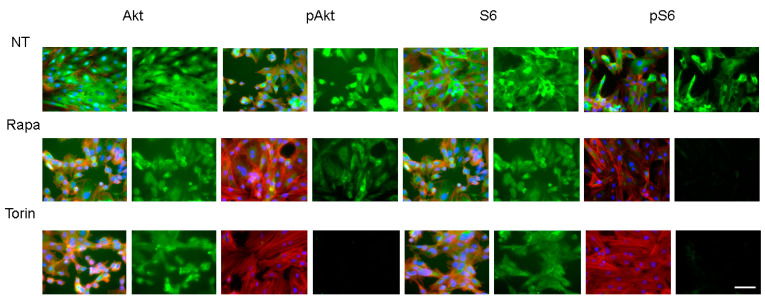
Representative confocal microscopy images of the immunolabeling of S6, p-S6, Akt, and p-Akt of the 2D H9c2 cells. 2D H9c2 cells treated or untreated (NT, non-treated) with 10 nM Rapa (Rapa) or Torin 1 (Torin) during a 6-day period were each immunostained for S6, p-S6, Akt, and p-Akt (green) without or with phalloidin (red) and DAPI (blue) in duplicate using fresh preparations (*n* = 5). Scale bar; 100 μm. The staining intensities are plotted in the lower panels. ** *p* < 0.01, *** *p* < 0.005.

**Figure 7 ijms-24-11459-f007:**
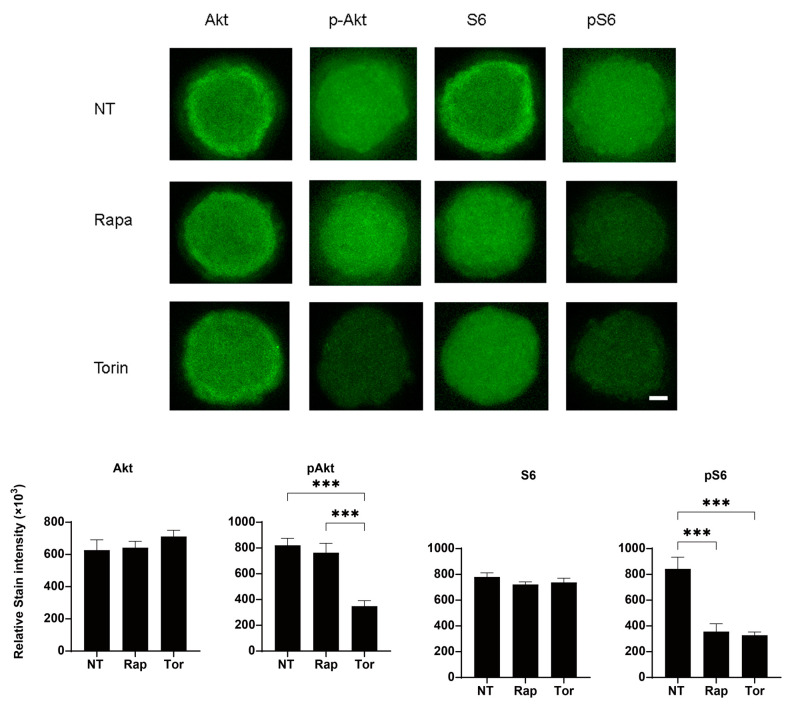
Representative confocal microscopy images of the immunolabeling of S6, p-S6, Akt, and p-Akt of the 3D H9c2 spheroids. 3D H9c2 spheroids treated or untreated (NT, non-treated) with 10 nM Rapa (Rapa) or Torin 1 (Torin) during a 3-day period were each immunostained for S6, p-S6, Akt, and p-Akt (green) in duplicate using fresh preparations (*n* = 5). Scale bar; 100 μm. The staining intensities are plotted in the lower panels. *** *p* < 0.005.

**Figure 8 ijms-24-11459-f008:**
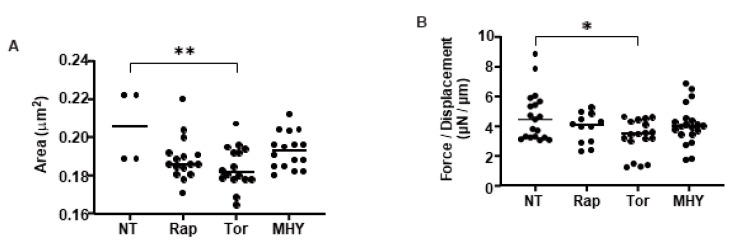
The physical characteristics of 3D H9c2 spheroids in the absence and presence of mTOR inhibitors or an mTOR stimulator. The mean sizes (μm) (**A**) and stiffness (μN/μm) (**B**), the force (μN) required to compress to their semidiameter (μm) during 20 s, of the 3D H9c2 spheroids were measured in the absence (NT, non-treated) or presence of the mTOR inhibitor, 10 nM Rapa (Rap), Torin 1 (Tor), or mTOR stimulator, 5 μM MHY-1485 (MHY) during 3 days in triplicate using fresh preparations (total *n*= 16), and those values were plotted. * *p* < 0.05, ** *p* < 0.01.

**Figure 9 ijms-24-11459-f009:**
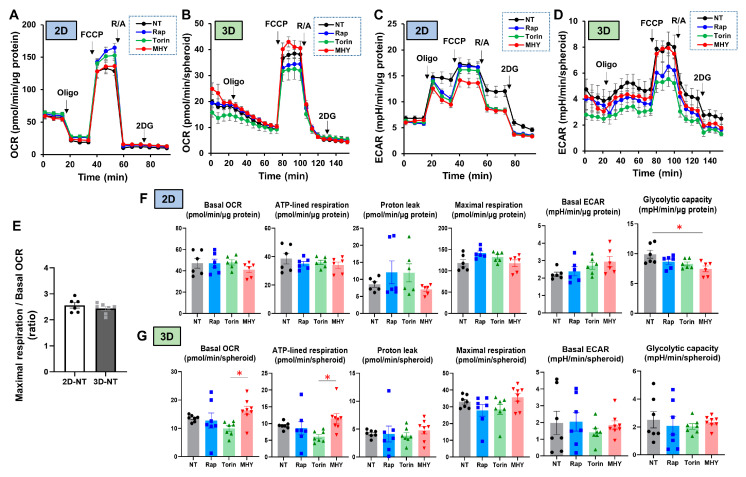
Assessment of metabolic functions in 2D and 3D H9c2 cells and their responses to mTOR modulators. A real-time metabolic function analysis using an XFe96 Extracellular Flux Analyzer of the 2D or 3D cultured H9c2 cells in the absence (NT, non-treated) or presence of mTOR inhibitor Rapamycin (Rap), Torin 1 (Tor) or mTOR stimulator MHY-1485 (MHY) on Day 3 in fresh preparations (*n* = 6–8). Panel (**A**): Measurement of OCR in 2D H9c2 cells. Panel (**B**): Measurement of OCR in 3D H9c2 spheroids. Panel (**C**): Measurement of ECAR in 2D H9c2 cells. Panel (**D**): Measurement of ECAR in 3D H9c2 spheroids. Panel (**E**): The ratio of maximal respiration to basal OCR in 2D and 3D conditions of H9c2 cells in the absence of mTOR modulators. Panel (**F**): Key indices of mitochondrial functions and glycolytic functions in 2D H9c2 cells. Panel (**G**): Key indices of mitochondrial functions and glycolytic functions in 3D H9c2 spheroids. Basal OCR = (the OCR at baseline)—(OCR with R/A), ATP-linked respiration = (the OCR at the baseline)—(the OCR with Oligo), Proton leak = (the OCR with Oligo)—(the OCR with R/A), Maximal respiration = (the OCR with FCCP)—(the OCR with R/A), Basal ECAR = (ECAR at the baseline)—(ECAR with 2DG), Glycolytic capacity = (ECAR with Oligo)—(ECAR with 2DG), OCR; oxygen consumption rate, ECAR; extracellular acidification rate, Oligo; oligomycin, FCCP; carbonyl cyanide p-trifluoromethoxyphenylhydrazone (FCCP), R/A; otenone/antimycin A, 2DG; 2-deoxyglucose. * *p* < 0.05.

**Figure 10 ijms-24-11459-f010:**
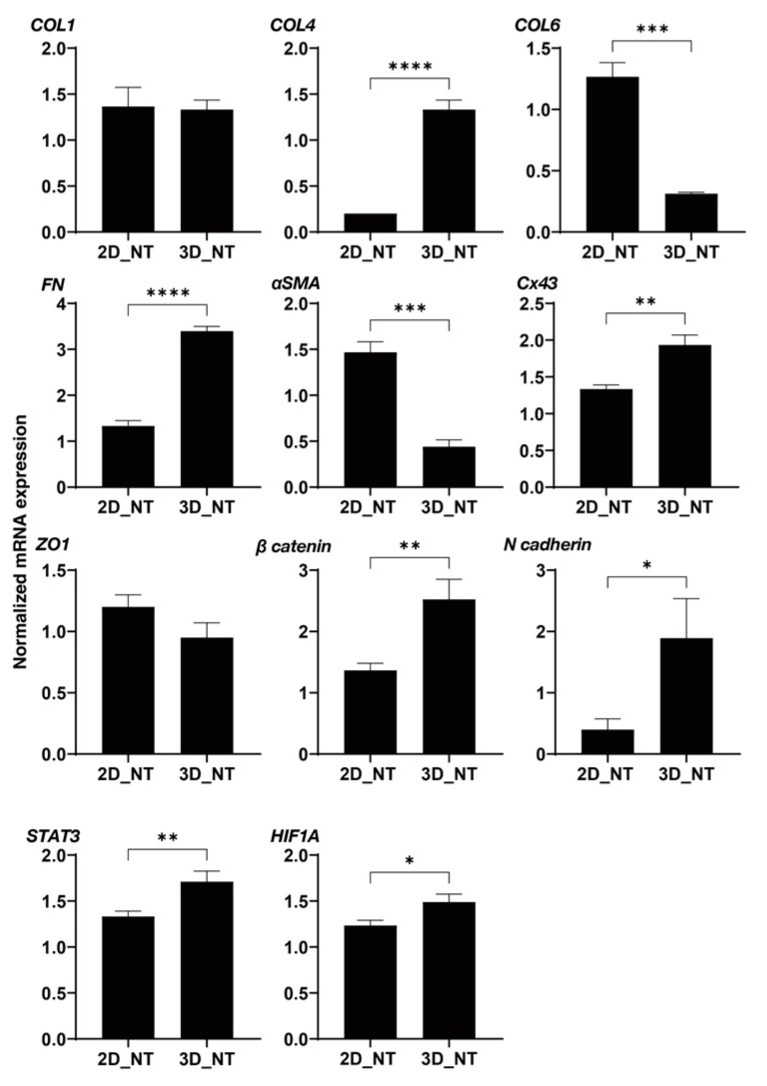
The mRNA expression of ECM molecules, gap junction-related molecules, STAT3, and HIF1 of 2D and 3D cultured H9c2 cells. qPCR analysis of ECM molecules (*COL1*, *COL4*, *COL6*, *FN,* and *a-SMA*), gap junction-related molecules (*ZO-1*, *β-catenin*, and *N-cadherin*), *STAT3*, and *HIF1* of non-treated (NT) 2D and 3D H9c2 cells with mTOR modulators in duplicate three different confluent six-well dishes (2D) or fifteen freshly prepared 3D spheroids (3D), and the resulting values were plotted. * *p* < 0.05, ** *p* < 0.01, *** *p* < 0.005, **** *p* < 0.001.

**Figure 11 ijms-24-11459-f011:**
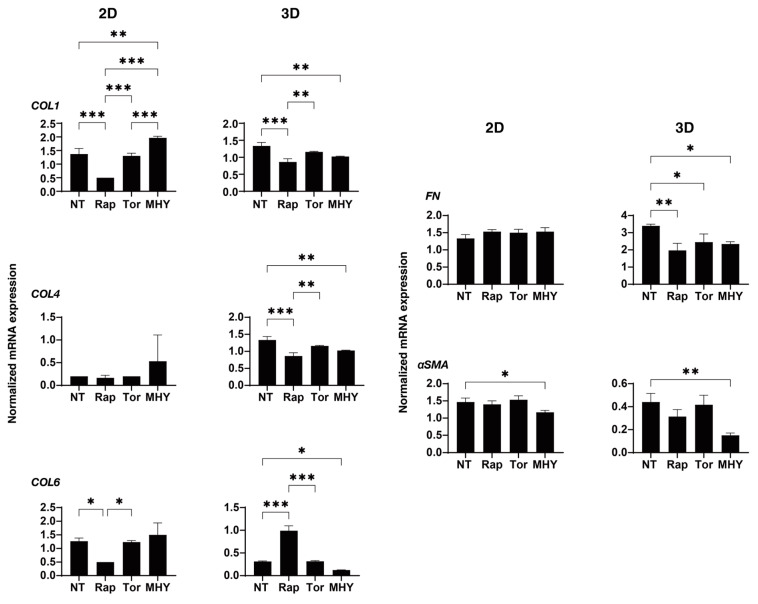
The mRNA expression of ECM molecules of 2D and 3D cultured H9c2 cells in the absence and presence of mTOR inhibitors or an mTOR stimulator. In the absence (NT, non-treated) or presence of mTOR inhibitor, 10 nM Rapa (Rap) or Torin 1 (Tor), or mTOR stimulator, 5 μM MHY-1485 (MHY) at Day 3, qPCR analysis of ECM molecules (*COL1*, *COL4*, *COL6*, *FN,* and *a-SMA*) of 2D and 3D H9c2 cells in duplicate 3 different confluent 6-well dishes (2D) or 15 freshly prepared 3D spheroids (3D), and those values were plotted. * *p* < 0.05, ** *p* < 0.01, *** *p* < 0.005.

**Figure 12 ijms-24-11459-f012:**
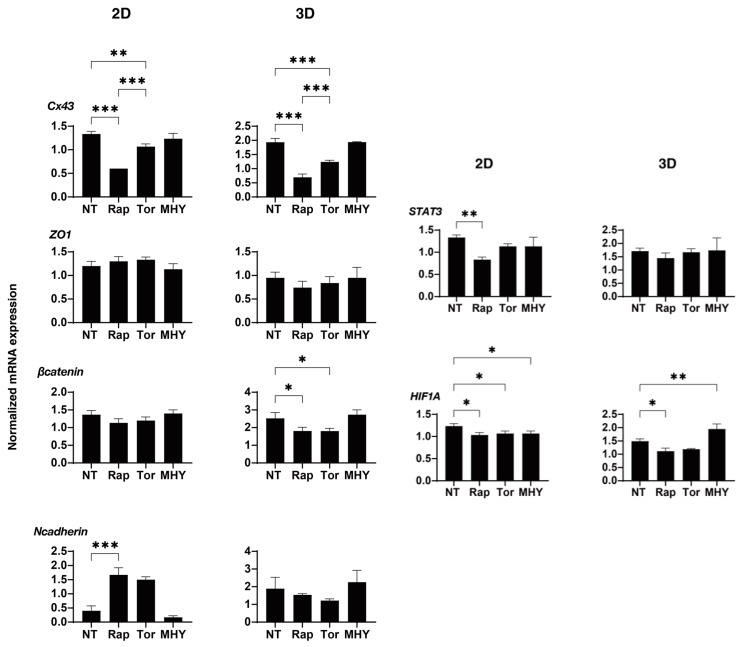
The mRNA expression of gap junction-related molecules, STAT3, and HIF1 of 2D and 3D cultured H9c2 cells in the absence and presence of mTOR inhibitors or an mTOR stimulator. In the absence (NT, non-treated) and presence of mTOR inhibitor, 10 nM Rapa (Rap) or Torin 1 (Tor), or mTOR stimulator, 5 μM MHY-1485 (MHY) on Day 3, qPCR analysis of gap junction-related molecules (ZO-1, β-catenin and N-cadherin), STAT3 and HIF1 of 2D and 3D H9c2 cells in duplicate three different confluent six-well dishes (2D) or fifteen freshly prepared 3D spheroids (3D), and those values were plotted. * *p* < 0.05, ** *p* < 0.01, *** *p* < 0.005.

**Table 1 ijms-24-11459-t001:** The number of cells within 2D and 3D spheroids at Day 3.

	NT	Rap	Tor	MHY
2D ^a^	36 ± 10.98	20.25 ± 4.19	8.75 ± 3.50	26 ± 1.15
3D ^b^	19,843 ± 2002	17,673 ± 2622	18,707 ± 2725	20,449 ± 2891

^a^ cell density: cell number/0.1 mm^3^. ^b^ total cells within: 3D spheroid.

## Data Availability

The data that support the findings of this study are available from the corresponding author upon reasonable request.

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
