# Peer review of "mTOR Inhibitors Modulate the Physical Properties of 3D Spheroids Derived from H9c2 Cells"

_ijms, 2023, doi:10.3390/ijms241411459_

Round 1
Reviewer 1 Report
The authors generated 3D spheroids from H9c2 cardiomyoblasts to establish an in vitro model for the local environment of cardiomyocytes. The spheroids were characterized using phase contrast and scanning electron microscopy, physical stiffness measurements, cellular metabolic analyses, and the expression of extracellular matrix (ECM) proteins and gap junction (GJ) related molecules.
The spheroids decreased in size and became stiffer over 3-4 days. In contrast to 2D planar cell cultures, which multiply, the 3D spheroids developed into a more mature form without any cell multiplication. qPCR analyses showed that the production of collagen4 (COL4), fibronectin (FN), connexin43 (CX43), β-catenin, N-cadherin, STAT3, and HIF1 molecules increased in the 3D spheroids, while the production of COL6 and α-smooth muscle actin (α-SMA) molecules decreased.
Treatment with rapamycin (Rapa) and Torin 1, mTOR inhibitors, reduced the size and gene expression of the 3D spheroids. Rapa and Torin1 also induced the formation of downsized and softened spheroids, respectively. The metabolic responses to mTOR modulators were different between the 2D and 3D cultures, with the mTOR activator MHY-1485 significantly increasing oxygen consumption for ATP synthesis in 3D spheroids only.
The findings suggest that the 3D H9c2 spheroids have different biological characteristics from 2D planar cultured H9c2 cells, and that these characteristics can be modulated by mTOR inhibitors. This suggests that 3D H9c2 spheroids may be a useful in vitro model that replicates the local environment of cardiomyocytes.
Essential Points:
1. Expression of collagen4 (COL4), fibronectin (FN), connexin43 (CX43), β-catenin, N-cadherin, STAT3, and HIF1 were analyzed using qPCR, to further establish 3D spheroids are an appropriate in vitro model for the local environment of cardiomyocytes, protein expression data is needed for some of the important proteins.
2. In figure 4 resolution of the images is not up to the mark, puncta need to show with arrows and quantification of the signal is required.
3. To establish mTOR signaling in 3D spheroids immunoblotting of S6, p-S6, Akt and p-Akt is necessary.
Minor points:
1. In table 1 some letters are overlapping
Author Response
Dear Editor,
Thank you very much for the constructive comments concerning our manuscript, " mTOR inhibitors modulate the physical properties of 3D spheroids derived from H9c2 cells”. We carefully checked all of the Reviewer comments and prepared a revised version of our paper that takes these comments into account. The changes are listed below. Specific changes within the manuscript are highlighted.
Reviewer 1 Reviewer 1 comments
The authors generated 3D spheroids from H9c2 cardiomyoblasts to establish an in vitro model for the local environment of cardiomyocytes. The spheroids were characterized using phase contrast and scanning electron microscopy, physical stiffness measurements, cellular metabolic analyses, and the expression of extracellular matrix (ECM) proteins and gap junction (GJ) related molecules.
The spheroids decreased in size and became stiffer over 3-4 days. In contrast to 2D planar cell cultures, which multiply, the 3D spheroids developed into a more mature form without any cell multiplication. qPCR analyses showed that the production of collagen4 (COL4), fibronectin (FN), connexin43 (CX43), β-catenin, N-cadherin, STAT3, and HIF1 molecules increased in the 3D spheroids, while the production of COL6 and α-smooth muscle actin (α-SMA) molecules decreased.
Treatment with rapamycin (Rapa) and Torin 1, mTOR inhibitors, reduced the size and gene expression of the 3D spheroids. Rapa and Torin1 also induced the formation of downsized and softened spheroids, respectively. The metabolic responses to mTOR modulators were different between the 2D and 3D cultures, with the mTOR activator MHY-1485 significantly increasing oxygen consumption for ATP synthesis in 3D spheroids only.
The findings suggest that the 3D H9c2 spheroids have different biological characteristics from 2D planar cultured H9c2 cells, and that these characteristics can be modulated by mTOR inhibitors. This suggests that 3D H9c2 spheroids may be a useful in vitro model that replicates the local environment of cardiomyocytes.
Essential Points:
- Expression of collagen4 (COL4), fibronectin (FN), connexin43 (CX43), β-catenin, N-cadherin, STAT3, and HIF1 were analyzed using qPCR, to further establish 3D spheroids are an appropriate in vitro model for the local environment of cardiomyocytes, protein expression data is needed for some of the important proteins.
Answer; Thank you for this comment. As suggested, to confirm qPCR analysis, immunoblots of 2D and 3D H9c2 samples treated N-cadherin antibody are now included in supplemental Figure 1 as a representative molecule of tight junction related factors.
- In figure 4 resolution of the images is not up to the mark, puncta need to show with arrows and quantification of the signal is required.
Answer; Thank you for this comment. As suggested, the resolutions of the images of Fig. 4 were improved and quantification is now included.
- To establish mTOR signaling in 3D spheroids immunoblotting of S6, p-S6, Akt and p-Akt is necessary.
Answer; Thank you for this comment. As suggested, immunolabeling images of 3D spheroid are now included.
Minor points:
- In table 1 some letters are overlapping
Answer; Thank you for this comment. As pointed out, I apologize for this careless mistake and thus this overlap was fixed.
Reviewer 2 comments
In this research, the authors prepared three-dimensional (3D) spheroids derived from H9c2 cardiomyoblasts by using 3D drop culture technique to establish an appropriate in vitro model for the local environment of cardiomyocytes and study how mTOR inhibitors modulate the physical properties of it. It was found that the mTOR inhibitors modulated biological aspects of the 3D H9c2 spheroids are quite different from that of 2D planar cultured H9c2 cells.
There are still some errors in this manuscript which need to be carefully corrected before publication. If the following issues are well-addressed, this reviewer believes that the essential contribution of this paper is important for understanding cardiac pathophysiology.
- The abstract is too long, which may prevent readers from getting the main findings of this paper. The authors should shorten the content of this section.
Answer; Thank you for this comment. As suggested, the abstract was changed to be more focused and shortened “To establish an appropriate in vitro model for the local environment of cardiomyocytes, three-dimensional (3D) spheroids derived from H9c2 cardiomyoblasts were prepared, and their morphological, biophysical phase contrast and biochemical characteristics were evaluated. 3D H9c2 spheroids were successfully obtained, the sizes of the spheroids decreased and they became stiffer during 3-4 days. In contrast to the cell multiplication that occurs in conventional 2D planar cell cultures, the 3D H9c2 spheroids developed into a more mature form without any cell multiplication being detected. qPCR analyses of the 3D H9c2 spheroids indicated that the production of collagen4 (COL4) and fibronectin (FN), connexin43 (CX43), b-catenin, N-cadherin, STAT3, and HIF1 molecules had increased and that the production of COL6 and a-smooth muscle actin (a-SMA) molecules had decreased as compared to 2D cultured cells. In addition, treatment with rapamycin (Rapa), an mTOR complex (mTORC) 1 inhibitor, and Torin 1, an mTORC1/2 inhibitor, resulted in significantly decreased cell densities of the 2D cultured H9c2 cells, but the size and stiffness of the H9c2 cells within the 3D spheroids were reduced with the gene expressions of several of the above several factors being reduced. The metabolic responses to mTOR modulators were also different between the 2D and 3D cultures. These results suggest that as unique aspects of the local environments of the 3D spheroids, the spontaneous expression of GJ-related molecules and hypoxia within the core may be associated with their maturation, suggesting that this may become a useful in vitro model that replicates the local environment of cardiomyocytes.”.
- Figure 1 is a too large figure which includes too many parts and cause page 4 only has a figure. The authors should break this figure into two parts or three parts.
Answer; Thank you for this comment. As suggested, Fig.1 contents were divided into three figures (Figs 1 -3).
- in line 210, the scale bar should be “100 μm“, which has a print error now.
Answer; Thank you for this comment. As pointed out, I apologize for this careless mistake and thus this was corrected.
- In line 255, it should be “Scale bar: 100 μm“. The authors should read the whole manuscript carefully to discover and correct these detailed errors.
Answer; Thank you for this comment. As pointed out, I apologize for this careless mistake and thus this was corrected. In addition, to avoid such kinds of careless mistakes, I carefully checked whole manuscript.
- The authors should correctly label the A to G panels in Figure 6. It is more reasonable that the author should properly divide this figure into two small figures because the current layout makes it difficult for readers to distinguish the labels in this figure.
Answer; We sincerely appreciate your pointing this out to us. We agree that the labels for the panels are lacking, the values were small, and the data for panel E was missing in the previous Figure 6. In the revised Figure 6, these problems have been amended.
- For Table 1 in lines 213 to 216 and Figure 7 in lines 334 to 357, the line number and the table/picture are overlapped together, the author should ensure that these two are not in editable mode.
Answer; Thank you for this comment. As pointed out, I apologize for this careless mistake and thus this overlap was fixed.
- In the Discussion section, the author should first summarize the main findings of this paper and focus on the discussion of the new findings of it.
Answer; Thank you for this comment. As suggested, we first summarized the main findings of this paper and then focused on the discussion of the new findings within the 1st paragraph of Discussion; “In the current investigation, 3D H9c2 spheroids were successfully produced by the same experimental protocol using the 384-hanging drop array plate as was described in our recent studies using mouse preadipocyte 3T3-L1 cells [27] as well as various other cells. The formation of 3D H9c2 spheroids and subsequent maturation as shown Fig. 1 closely resembled those for 3T3-L1 cells [27]. Interestingly, the results of qPCR analyses suggested that GJ related molecules, which are essentially required for the cardiac functional syncytium [41], were spontaneously expressed upon 3D H9c2 spheroid formation. In addition, since it has been shown that an O2 gradient exists within the 3D spheroids, similar to the cardiac wall, the significantly higher expression of the HIF1 gene may also suggest the existence of such an O2 gradient within our 3D H9c2 spheroid, although changes in protein levels of HIF1 and its downstream protein were not measured. In addition, mTOR modulators induced different effects toward cellular proliferations as well as the cellular metabolic states of 2D and 3D cultured H9c2 cells. Therefore, these collective current results rationally suggest that our newly developed in vitro 3D H9c2 spheroid model could be more applicable than conventional 2D cultured models for use in research in the fields of cardiac physiology as well as pathology.”.
- Authors need to unify the format of all references, such as whether to use both the starting and ending page numbers (Ref. 44), and whether to abbreviate the journal name (Ref. 30, 31, 33, 47, 49, 50, 53).
Answer; Thank you for this comment. As pointed out, some of the references are without both starting and ending page numbers. Nevertheless, references were edited by newest version of endnote software to choose MDPI style. In addition, we also carefully checked pointed refs by PUBMED, and those were missing ending pages for example, Sci rep papers.

Reviewer 2 Report
In this research, the authors prepared three-dimensional (3D) spheroids derived from H9c2 cardiomyoblasts by using 3D drop culture technique to establish an appropriate in vitro model for the local environment of cardiomyocytes and study how mTOR inhibitors modulate the physical properties of it. It was found that the mTOR inhibitors modulated biological aspects of the 3D H9c2 spheroids are quite different from that of 2D planar cultured H9c2 cells.
There are still some errors in this manuscript which need to be carefully corrected before publication. If the following issues are well-addressed, this reviewer believes that the essential contribution of this paper is important for understanding cardiac pathophysiology.
- The abstract is too long, which may prevent readers from getting the main findings of this paper. The authors should shorten the content of this section.
- Figure 1 is a too large figure which includes too many parts and cause page 4 only has a figure. The authors should break this figure into two parts or three parts.
- in line 210, the scale bar should be “100 μm“, which has a print error now.
- In line 255, it should be “Scale bar: 100 μm“. The authors should read the whole manuscript carefully to discover and correct these detailed errors.
- The authors should correctly label the A to G panels in Figure 6. It is more reasonable that the author should properly divide this figure into two small figures because the current layout makes it difficult for readers to distinguish the labels in this figure.
- For Table 1 in lines 213 to 216 and Figure 7 in lines 334 to 357, the line number and the table/picture are overlapped together, the author should ensure that these two are not in editable mode.
- In the Discussion section, the author should first summarize the main findings of this paper and focus on the discussion of the new findings of it.
- Authors need to unify the format of all references, such as whether to use both the starting and ending page numbers (Ref. 44), and whether to abbreviate the journal name (Ref. 30, 31, 33, 47, 49, 50, 53).
Author Response

(The authors gave the same response as above.)

Round 2
Reviewer 2 Report
After the suggestions were all seriously considered by the authors, the reviewer believes this version of the manuscript is suitable to be published.